# Diagnostic Classification of Cases of Canine Leishmaniasis Using Machine Learning

**DOI:** 10.3390/s22093128

**Published:** 2022-04-20

**Authors:** Tiago S. Ferreira, Ewaldo E. C. Santana, Antônio F. L. Jacob Junior, Paulo F. Silva Junior, Luciana S. Bastos, Ana L. A. Silva, Solange A. Melo, Carlos A. M. Cruz, Vivianne S. Aquino, Luís S. O. Castro, Guilherme O. Lima, Raimundo C. S. Freire

**Affiliations:** 1Graduating Program in Computation Engineering and Systems, State University of Maranhão, São Luís 65690-000, Brazil; tiagoferreira3@aluno.uema.br (T.S.F.); ewaldosantana@professor.uema.br (E.E.C.S.); antoniojunior@professor.uema.br (A.F.L.J.J.); 2Graduating Program in Animal Sciences, State University of Maranhão, São Luís 65690-000, Brazil; bastos.luciana@discente.ufma.br (L.S.B.); anasilva1@professor.urma.br (A.L.A.S.); 3Graduating Program in Animal Health Defense, State University of Maranhão, São Luís 65690-000, Brazil; solagemelo@professor.uema.br; 4Graduation Program in Electrical Engineering, Federal University of Amazonas, Manaus 69067-005, Brazil; carlosamcruz@ufam.edu.br (C.A.M.C.); vivisaquino@gmail.com (V.S.A.); luis.oliveira@sidia.com (L.S.O.C.); 5Graduation Program in Electrical Engineering, Federal University of Maranhão, São Luís 65690-000, Brazil; guilhermeolima2010@gmail.com; 6Graduation Program in Electrical Engineering, Federal University of Campina Grande, Campina Grande 58428-830, Brazil; rcsfreire@dee.ufcg.edu.br

**Keywords:** machine learning, classification, logistic regression, canine visceral leishmaniasis

## Abstract

Proposal techniques that reduce financial costs in the diagnosis and treatment of animal diseases are welcome. This work uses some machine learning techniques to classify whether or not cases of canine visceral leishmaniasis are present by physical examinations. For validation of the method, four machine learning models were chosen: K-nearest neighbor, Naïve Bayes, support vector machine and logistic regression models. The tests were performed on three hundred and forty dogs, using eighteen characteristics of the animal and the ELISA (enzyme-linked immunosorbent assay) serological test as validation. Logistic regression achieved the best metrics: Accuracy of 75%, sensitivity of 84%, specificity of 67%, a positive likelihood ratio of 2.53 and a negative likelihood ratio of 0.23, showing a positive relationship in the evaluation between the true positives and rejecting the cases of false negatives.

## 1. Introduction

Proposal techniques that reduce financial costs in the diagnosis and treatment of animal diseases are welcome. Among the poorest peoples in several parts of the world, there are one of the most severe forms of leishmaniasis, the visceral leishmaniasis (VL), also known as kalazar. VL is a life-threatening disease caused by Leishmania parasites, which are transmitted by female sandflies. VL causes fever, weight loss, spleen and liver enlargement, and, if not treated, death. People with both visceral leishmaniasis and HIV are difficult to cure [1]. The World Health Organization estimates that from 700,000 to 1,000,000 new VL cases occur annually [2]. VL is present in 88 countries, 22 of which are in the Americas. It is estimated that Brazil handles 90% of VL cases in Latin America [2]. Catão. R.C. [3] says that there is a stable interrelationship between the pathogen, vectors and people (infected and susceptible) with the geographic space. With leishmaniasis, mappings can help to understand the dynamics of transmission and the behavior of vectors. Injuries may be confined to one location or reach larger areas. Therefore, knowledge of spatial patterns in the occurrence of the disease becomes important for case surveillance [4].

The VL diagnoses are made by laboratory exams like the ELISA (enzyme-linked immunosorbent assay) test. These tests are sometimes very expensive for most residents in poor countries demonstrating the need for technologies that can reduce cost. In pursuing this aim, machine learning techniques appear as one of the most efficient methods in detecting VL in infected dogs.

Machine learning (ML) techniques employ the principle of induction by induction, getting results and extrapolations from a particular set of examples [5,6,7]. The ML system can be defined as a multi-component system, with an interface, learning algorithm, data, infrastructure and hardware. The learning algorithm is classified in two major categories: Supervised and unsupervised. In supervised learning, knowledge of the external environment is presented by sets of examples as desired input and output, in which the ML algorithm extracts the knowledge representation from these examples. The aim is that the generated representation can produce correct outputs for new inputs not presented [5,6,7]. With unsupervised learning, the model will not receive the desired output. The goal is for the machine to extract information from the input variables in order to separate them into different classes [8]. Unsupervised learning is the most widely used type of machine learning [9] and regression models (supervised) are the most used predictive model types, and among them, logistic regression analysis is used for dichotomous outputs. Logistic regression accounts for or predicts values of a single result variable with information from one or more explanatory variables and can classify an observation into one of two or more classes [10]. Logistic regression is one of the most used analytical tools in social and natural sciences [11].

Several studies have used machine learning to diagnose canine diseases. Larius, G. et al. [12] developed a method for the diagnosis of canine visceral leishmaniasis based on Fourier-Transform Infrared Spectroscopy (FTIR spectroscopy) and machine learning, in which canine blood sera from twenty uninfected dogs, twenty Leishmania infantum and eight dogs infected with *Trypanosoma evansi* were analyzed. They used principal component analysis with machine learning algorithms and archived over 85% in diagnosing true positives. Reagan, KL et al. [13] also applied machine-learning techniques to aid in the diagnosis of Canine Hypoadrenocorticism (CH) using screening diagnosis by complete blood count and serum chemistry panel. The database used was 908 control dogs with suspected CH and 133 dogs with confirmed CH. A driven tree algorithm was trained and tested to assess performance, with a sensitivity of 96.3%, and a specificity of 97.2%. A lymph node parasite load prediction model from clinical data in dogs with visceral leishmaniasis by artificial neural networks and machine learning was presented in [14]. In this study, 55 (fifty-five) dogs from seven regions of the states of Bahia, Minas Gerais, São Paulo and Distrito Federal, with 35 infected dogs and twenty control dogs, archived accuracy of 78% in the analyses performed. In the research carried out in [15], four machine learning algorithms were used to predict the diagnosis of Cushing’s syndrome, using structured clinical data from the VetCompass program in the UK. Cushing’s syndrome, which is an endocrine disease in dogs, negatively affects the quality of life of affected dogs. Machine learning methods could classify the recorded Cushing syndrome diagnoses, with a predictive result for regression with a sensitivity of 0.71 and a specificity of 0.82. We can notice that in these works all researchers used some kind of laboratory exam.

In this work, we propose the use of machine learning methods to make a machine that predicts if a certain animal has or does not have canine visceral leishmaniasis based only on physical examination of it. For that, four machine learning algorithms were used and the best one was chosen in the classification case of VL. Data were used from canine clinical exams realized in a certain region of the state of Maranhão, Brazil, to be used in the models.

This work is divided into three more sections besides this introduction. In Section 2, the materials and methods used in the work are discussed, in Section 3 the results are presented, and in Section 4 the final considerations are given.

## 2. Materials and Methods

According to Bassert, J.M. et al. [16] “History and physical examination are the first steps in the technician’s observation of any patient or group of patients. The information obtained from these processes serves as the basis for all subsequent assessments and interventions. It is essential that veterinary technicians can get complete and accurate historical information in the assessments of each patient and group. Similarly, good physical examination skills allow the quick identification of significant problems, followed by appropriate therapeutic measures”. The physical examination includes a professional assessment of the patient’s health and well-being.

In this work, the database was created from existing clinical examination records on 340 (three hundred and forty) dogs (cases: *n* = 177, non-cases: *n* = 163). We got seventeen variables that describe the dog’s characteristics, as seen in Table 1. These variables, according to the veterinaries, are variables that they observe in a first view of the animal suspected of having VL: Sex, presence of ectoparasites, nutrition, lymph nodes, mucosal color, bleeding, coat, muzzle and/or ear injury, nails, presence of skin lesion, depigmentation, alopecia, eye secretion, blepharitis, proximity to the forest and the ELISA (enzyme-linked immunosorbent assay) test results. With that information from veterinaries, our initial step was to use these variables to train the models. In Table 1 we also can see the *p* = value of a correlation test between the levels of the variables.

The ELISA test is a quantitative serological method making up a tool used both for analysis of clinical suspicion and for confirming the diagnosis of leishmaniasis. Confirmation occurs through the detection of immunoglobulinG (IgG) in the serum of suspected dogs. This exam is chosen because of its specificity and sensitivity [17,18]. The ELISA test’s performance is related not only to the type of antigen used but also to the clinical state manifested by the dog [19]. This test is considered the golden standard test in the diagnosis of leishmaniasis and it is the confirmatory test recommended by the Brazilian Ministry of Health [20]. In this work, the results of the ELISA (positive or not) test are used as the dependent (target) variable.

Data collection was performed out in certain regions of the west of the state of Maranhão (1°59′–4°00 S and 44°21′–45°33′ W), which is low in the Human Development Index (HDI) [21] (Figure 1).

### 2.1. Model Selection and Variable Selection

The target variable, ELISA test results, is dichotomous and Logistic Regression (LR) appears as a good choice for the learning algorithm [22,23,24,25]. Four algorithms were tested to choose the best model: Support Vector Machine (SVM), K-Nearest Neighbor (KNN), Naïve Bayes (NB) and LR, as well as the K-nearest neighbor classifier, which is based on the characteristic of the k-nearest neighbor of a new point (sample) to classify it. In this work, the best results were achieved with k = 10.

Naïve Bayes classifier is based on the assumption of independence between the variables of the problem. The NB model performs a probabilistic classification of an unclassified sample to put it in the most likely class.

Support vector machine is a high-performance model for nonlinear problems, not biased by outliers and not sensitive to them. It includes Support Vector Classification (SVC) and Support Vector Regression (SVR) [26].

For each model, we applied a recursive feature elimination with cross-validation as a preprocessing step [27,28] to select the best variables. For the SVM model, the excluded variables were age, lymph nodes and eye secretion. For LR, NB and KNN model the variables excluded were age, condition and depigmentation. The algorithms were trained and tested with the dataset containing only the best variables.

Regression models are one of the most important statistical tools in the statistical analysis of data for modelling relationships between variables. These models aim to detect the relationship between one or more explanatory variables, and response, or dependent variables. One of the particular cases of generalized logistic models is the one in which the response variable has only two categories of dichotomized values (0 or 1) [10].

Logistic regression aims to model, from a set of observations, the logistic relationship (probability distribution) between a dichotomous response variable and a series of numerical explanatory variables, which can be continuous, discrete and categorical [10,22]. The idea is to use the logistic expression given by:y = (1 + e^−z^)^−1^(1)
where z = a_0_ + a^T^X, X is an m × n matrix containing m examples with n features, y is an m × 1 array of 0 and 1, and a is an m × 1 vector containing the parameters of the system, which will be inferred by the learning algorithm. This inference is done by an interactive task aiming to minimize the error between the actual values and the inferred values of y. After obtaining the parameter vector, a, one can infer a value for a new sample. The classification is made in the following way: The learning algorithm will, for each example, determine a number by equation (1), which represents the probability of y = 1, and if this number is equal or greater than 0.5 will put y = 1, or 0 otherwise. With the parameter vector a, we can assign a number for each new dog feature vector shown to the system.

For example, we can assign the 240 × 14 feature matrix (after variables selection), X, containing the values of the 14 features for each one of the 240 dogs and a 240 × 1 vector, y, containing the values 0 or 1 if the dog does not have or has the disease. We separate this sample into two parts: 80% for training and 20% for test. For the training set, we present the learning algorithm with a 192 × 14 matrix and a corresponding 192 × 1 vector y. In this learning phase, the algorithm will estimate the parameters a_i_, i = 0, …, 14. With the estimated vector a, we can get z and put it in Equation (1) to estimate the values of the y for each sample in the test set. With the inferred and actual values, we can get the confusion matrix to get the metrics explained in the next section. In this work, after the training phase, we got a = [0.00401544, −0.3851127, 0.25599153, −0.05313171, 0.54447301, 0.36488774, −0.21396936, 0.15184775, 0.28558144, −0.11643821, 0.47422156, 0.398408060, −0.34161375, −1.24370334] ^T^ and a_0_ = 0.30852909.

### 2.2. Diagnostic Test

Diagnosing a disease is a delicate matter because the lives of patients are at stake, whether they are humans, dogs or even plants. The tools used in the diagnostic process are tests based on measurements made on patients, whether quantitative or qualitative, called clinical tests or diagnostic tests [23,24,25].

These tools have become so important and widespread that there are large industries and laboratories entirely dedicated to the production of increasingly accurate, rapid and inexpensive diagnostic tests. Tests can be misleading, especially where there may be a problem with a biological system. Before a test is used as an aid in the diagnosis of a certain disease, its potential for error must be evaluated [23,24,25].

Technological proposals to reduce the financial costs of treating diseases and the use of general laboratory tests are of great interest. These technologies act like screening tests leaving laboratory tests to be performed only on beings with a high probability of disease presence. The machine learning techniques can help identify sick individuals with a reasonable statistical probability of true positives.

Yang Xin et al. [26] say: “The evaluation model is a very important part of the machine-learning mission”. In this work, we follow their steps to evaluate our proposal, using the metrics obtained from the confusion matrix. The confusion matrix is shown in Table 2.

Further, the following metrics can be calculated from the confusion matrix [26]:

Accuracy: (TN + TP)/(TN + FP + FN + TP). This measures the fraction of correct predictions.

Sensitivity or Recall: (TP)/(TP + FN). This measures the ability of the test to correctly identify individuals who have the disease. It measures the probability of the test getting a positive result given that the true condition is present. This is the most important metric in screening because a negative result in a test with high sensitivity is useful for excluding the existence of the condition.

Specificity: TN/(TN + FP). This is the ability of the test in correctly identify individuals who do not have the disease;

The Positive Predictive Value or Precision: TP/(TP + FP). This measures the probability of the dog having the disease knowing that the test result is positive.

The Negative Predictive Value: TN/(TN + FN). This measures the probability of the dog has not had the disease knowing that the test result is negative.

The Positive Likelihood Ratio (LR+): Sensitivity/(1 − Specificity). This shows that for a value greater than 1 (one), the positive test is more likely to occur in dogs with the disease than in those without the disease;

The Negative Likelihood Ratio (LR−): (1 − Sensitivity)/Specificity. This shows that for a value greater than 1 (one), the negative test is more likely to occur in dogs with the disease than in those without the disease.

The Area Under Curve (AUC): from the Receiver Operating Characteristic (ROC) curve: This is performed to identify how good the model developed is at distinguishing between two parameters, the true positive rate and the false negative rate. Models with 100% correct predictions have an AUC of 1.

### 2.3. Canine Visceral Leishmaniasis

Leishmaniasis belongs to the group of diseases caused by a parasitic protozoan of the genus Leishmania which is transmitted to humans and other various mammals through the bite of females of a hematophagous insect dipterans of the *Psychodidae* family, subfamily *Phlebotominae*, known generically as sandflies, playing the role of a vector in the disease cycle [2,29,30]. The World Health Organization has included leishmaniasis as one of the six most important diseases in the world. Even included in this list, leishmaniasis is considered a neglected disease. It is related to the poverty of people with deteriorating housing and bad sanitation conditions and is common in regions with a low economic development index.

Furtado, A. S. et al. [31] say that Maranhão showed an expansion of cases of human leishmaniasis in the period from 2000 to 2009. From 1999 to 2005, the state led the number of confirmed cases of the disease in Brazil. In the year 2019, according to data from the Notification System of the Health Surveillance Secretariat of the Ministry of Health, 430 confirmed cases of visceral leishmaniasis in humans were reported [1], which shows the importance of research in early detection of main vectors of disease spread.

## 3. Results and Discussions

The database was split into eighty per cent for training and twenty per cent for testing. The training set was used to determine the parameters of the learning algorithm and the test set was used to validate the models by the metrics described above. Table 3 shows the results for the models used.

From Table 3, one can see that the LR model got better results than the others did, for instance, accuracy of 75%, sensitivity of 84% and negative predictive value of 83%. These are good results because they assure us great security in that the dog predicted as not having the disease does not actually have it; then it is not necessary to carry out a laboratory exam on it. A positive predictive value of 0.69 means that we have approximately 70% certainty that the dogs tested as positive really have the disease. The LR+ equal to 2.53 means that a dog with the disease is 2.53 times more likely to have a positive test than one without the disease. The LR− equal to 0.23 means that a dog without the disease is, approximately, four (0.25) times more likely to test negative than those with the disease. Thus, the logistic regression model shows a good ability in rejecting false negatives.

The AUC of 0.77 (Figure 2) shows the test’s discriminatory ability to distinguish between dogs with and without the disease.

From those results, one can observe that the logistic regression model can act as an efficient screening method for dogs with canine visceral leishmaniasis based only on their visualization and thus reducing the cost in laboratory exams.

As an attempt to understand the type of correctly and not correctly classified samples, for the LR model, we get the descriptive characteristics of each variable for the four classifications: TN, FP, FN and TP.

In Table 4 we have the confusion matrix. One can see that this model got five false negative and 12 false positive samples.

Table 5 shows the samples classified as false negative. One can see that 100% of the samples present the following characteristics: Normal mucosal color, no bleeding, augmented nails, no presence of skin lesion, no eye secretion and no blepharitis.

Table 6 shows the samples classified as false positive. One can see that 100% of the samples present the following characteristics: No bleeding and no eye secretion.

Table 7 shows the samples classified as true negative. One can see that samples possess the following characteristics: No presence of skin lesion and no eye secretion.

Table 8 shows the samples classified as true positive. One can see that a majority of true positive samples have no presence of ectoparasites, enlarged lymph nodes, no presence of bleeding, no presence of skin lesion, no eye secretion, no blepharitis and no proximity to the forest.

## 4. Final Considerations

In this work, four machine learning models were tested as an initial method in veterinary care to identify dogs with canine visceral leishmaniasis based only on visual inspection of the animal. For that, we got clinical dates from 340 dogs with eighteen variables. These variables were chosen based on veterinary professionals’ experiences and for each model the best variables were selected to predict the results. The models tested were logistic regression, support vector machine, K-nearest neighbor, and Naïve Bayes. The logistic regression model, using fourteen variables after the variable selection procedure, got the best metrics: Accuracy of 75%, sensitivity of 84%, specificity of 67%, positive likelihood ratio of 2.53 and negative likelihood ratio of 0.23. This model enables cost reduction in this type of care and can become a useful tool to screen this disease, contributing to the improvement of urban public health.

## Figures and Tables

**Figure 1 sensors-22-03128-f001:**
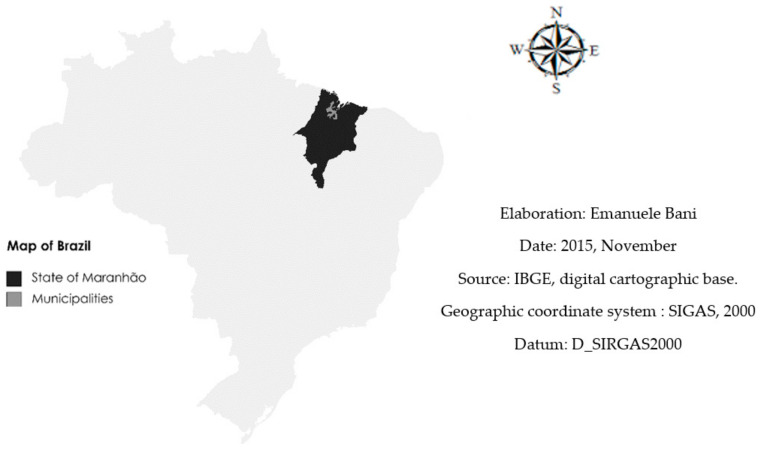
Region of Maranhão, Brazil, where the data were collected.

**Figure 2 sensors-22-03128-f002:**
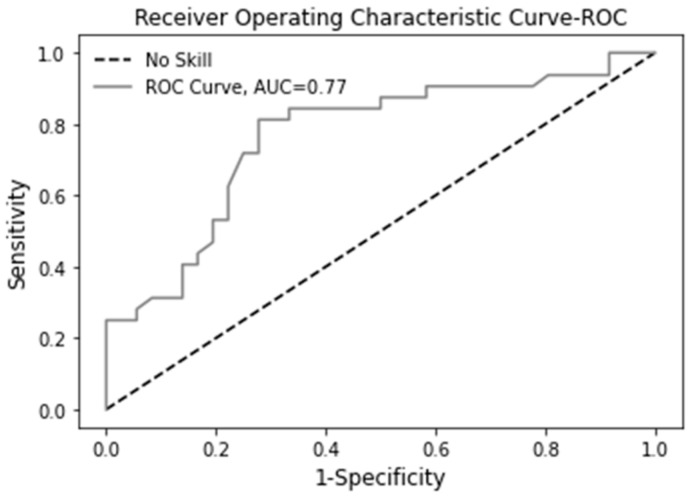
ROC curve for applying the LR model on the test set.

**Table 1 sensors-22-03128-t001:** Descriptive statistics and univariable associations of features included in machine learning prediction of the canine visceral leishmaniasis (cases: *n* = 177; non-cases: *n* = 163).

Variable	Category	Non-Cases	Cases	*p*-Value
Sex	Female	74	74	0.505
	Male	89	103	
Age (months)	Mean/standard deviation	34.39/30.8	44.03/36.45	0.009
Condition	Apathetic	19	27	0.346
	Active	144	150	
Presence of ectoparasites	No	127	151	0.078
	Yes	36	26	
Nutrition	Normal	119	113	0.058
	Thin	41	53	
	Skinny	3	11	
Lymph nodes	Normal	25	27	0.983
	Enlarged	138	150	
Mucosal color	Normal	121	123	0.332
	Pale	42	54	
Bleeding	No	156	162	0.118
	Yes	7	15	
Coat	Normal	87	65	0.007
	Regular	44	70	
	Bad	32	42	
Muzzle and/or ear injury	No	133	118	0.002
	Yes	30	177	
Nails	Augmented	127	100	<0.001
	onychogryphosis	36	77	
Presence of skin lesion	No	153	161	0.314
	Yes	10	16	
Depigmentation	No	162	177	0.479
	Yes	1	0	
Alopecia	No	116	89	<0.001
	Yes	47	88	
Eye secretion	No	159	166	0.115
	Yes	4	11	
Blepharitis	No	145	157	0.94
	Yes	18	20	
Proximity to the forest	No	77	141	<0.001
	Yes	86	36	

**Table 2 sensors-22-03128-t002:** Indications of the confusion matrix.

	Predicted as Negative	Predicted as Positive
Labeled as Negative	True Negative (TN)	False Positive (FP)
Labeled as Positive	False Negative (FN)	True Positive (TP)

**Table 3 sensors-22-03128-t003:** Test Metrics of the models tested. One can see that LR got the best metrics.

	NB	KNN	SVM	LR
Accuracy	0.63	0.63	0.69	0.75
Sensitivity (Recall)	0.56	0.56	0.84	0.84
Specificity	0.69	0.69	0.56	0.67
Positive Predictive Value	0.62	0.62	0.63	0.69
Negative Predictive Value	0.64	0.64	0.80	0.83
LR+	1.84	1.84	1.90	2.53
LR−	0.63	0.63	0.28	0.23
AUROC	0.71	0.71	0.70	0.77

**Table 4 sensors-22-03128-t004:** Confusion matrix.

	Predicted as Negative	Predicted as Positive
Labeled as Negative	24	12
Labeled as Positive	5	27

**Table 5 sensors-22-03128-t005:** Frequency of the variables for the false negatives samples.

Variable	Category	Frequency (%)
Sex	Female	20
	Male	80
Presence of ectoparasites	No	80
	Yes	20
Nutrition	Normal	60
	Thin	40
	Skinny	0
Lymph nodes	Normal	20
	Enlarged	80
Mucosal color	Normal	100
	Pale	0
Bleeding	No	100
	Yes	0
Coat	Normal	20
	Regular	60
	Bad	20
Muzzle and/or ear injury	No	80
	Yes	20
Nails	Augmented	100
	onychogryphosis	0
Presence of skin lesion	No	100
	Yes	0
Alopecia	No	80
	Yes	20
Eye secretion	No	100
	Yes	0
Blepharitis	No	100
	Yes	0
Proximity to the forest	No	20
	Yes	80

**Table 6 sensors-22-03128-t006:** Frequency of the variables for the false positive samples.

Variable	Category	Frequency (%)
Sex	Female	58
	Male	42
Presence of ectoparasites	No	0.92
	Yes	0.08
Nutrition	Normal	83
	Thin	17
	Skinny	0
Lymph nodes	Normal	8
	Enlarged	92
Mucosal color	Normal	75
	Pale	25
Bleeding	No	100
	Yes	0
Coat	Normal	50
	Regular	30
	Bad	20
Muzzle and/or ear injury	No	83
	Yes	17
Nails	Augmented	58
	onychogryphosis	42
Presence of skin lesion	No	83
	Yes	17
Alopecia	No	42
	Yes	58
Eye secretion	No	100
	Yes	0
Blepharitis	No	92
	Yes	8
Proximity to the forest	No	92
	Yes	8

**Table 7 sensors-22-03128-t007:** Frequency of the variables for the true negatives samples.

Variable	Category	Frequency (%)
Sex	Female	42
	Male	58
Presence of ectoparasites	No	75
	Yes	25
Nutrition	Normal	83
	Thin	17
	Skinny	0
Lymph nodes	Normal	8
	Enlarged	92
Mucosal color	Normal	67
	Pale	33
Bleeding	No	92
	Yes	8
Coat	Normal	62
	Regular	21
	Bad	17
Muzzle and/or ear injury	No	96
	Yes	4
Nails	Augmented	96
	onychogryphosis	4
Presence of skin lesion	No	0
	Yes	100
Alopecia	No	92
	Yes	8
Eye secretion	No	100
	Yes	0
Blepharitis	No	96
	Yes	4
Proximity to the forest	No	21
	Yes	79

**Table 8 sensors-22-03128-t008:** Frequency of the variables for the true positive samples.

Variable	Category	Frequency (%)
Sex	Female	30
	Male	70
Presence of ectoparasites	No	93
	Yes	7
Nutrition	Normal	67
	Thin	22
	Skinny	11
Lymph nodes	Normal	12
	Enlarged	88
Mucosal color	Normal	63
	Pale	37
Bleeding	No	89
	Yes	11
Coat	Normal	26
	Regular	37
	Bad	37
Muzzle and/or ear injury	No	63
	Yes	37
Nails	Augmented	44
	onychogryphosis	56
Presence of skin lesion	No	85
	Yes	15
Alopecia	No	40
	Yes	60
Eye secretion	No	85
	Yes	15
Blepharitis	No	93
	Yes	7
Proximity to the forest	No	97
	Yes	3

## Data Availability

The data used in this work are not available for consultation on the website, being the property of the Graduating Program in Animal Sciences of the State University of Maranhão.

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
