# Peer review of "Diagnostic Classification of Cases of Canine Leishmaniasis Using Machine Learning"

_sensors, 2022, doi:10.3390/s22093128_

Round 1

Reviewer 1 Report

This paper uses logistic regression to classify positive cases of canine visceral leishmaniasis. A sample of 340 animals was selected and then diagnosed by physical examination and also using ELISA (a very sensitive clinical test). A model was constructed using the variables collected in the physical examination and the ELISA serological test results. The model has decent prediction metrics, but the paper fails to explain if any other models were used as a comparison, or to describe the process of model selection. It would be interesting to see the following edits:

1) Evidence to support the fact that ELISA test results can be used to construct the response variable, i.e. evidence of its sensitivity.

2) A description of why the final model was chosen: are there any other models? are there variables that were included first and then eliminated? Did the authors compare several options for the model? If so, it would be nice to see a comparison table in the paper, and why they choose the final model.

3) From the ROC curve results, it would be interesting to see which are the type of cases that were misclassified. This could be useful to define future work and suggest ways to improve future models like this one. This can be done by describing the most common profile that was misclassified, and also the most common profile that was correctly classified. 

4) The results can be described in terms of the model. How can the model be used to predict a positive case? what is the final model? The authors are not including the coefficients nor the details on how to interpret them.

4) Writing can be improved. There are many typos and grammar mistakes in the text and sometimes is hard to understand the point. It would be very good if the authors could send the paper for a professional English revision. 

Author Response

In the attached file we insert the answers to the questions and the necessary changes, for better visualization we insert the article at the end of the text. 

Reviewer 2 Report

From a scientific point of view, the article does not bring anything new. The authors used a well-known machine learning method to identify cases of canine visceral leishmaniasis. Classification was done based on the parameters given in Table 1.
It would be more interesting to know how these parameters were selected - probably experimentally. It would be good to indicate the effectiveness of the classifier (Accuracy). How the results of the experiment were affected by the imbalance of the input data. Admittedly the imbalance is small, but was it studied?
The paper is applied, showing the use of ML to solve practical problems - veterinary here. Formally, the metrics that evaluate the classifier are not described by mathematical formulas.

Author Response

(The authors gave the same response as above.)

Reviewer 3 Report

Dear authors

I sincerely hope my comments are taken as constructive critcism only.  The manuscript has a great clinical significance. As the disorder is very common in poverty struck areas where cost of laboratory testing is prohibitive of reaching the final diagnosis, machine learning model would increase the confidence of veterinary practitioner in reaching the final clinical diagnosis of the disorder in dogs as important reservoirs for humans.

The manuscript needs insignificant corrections (see below)

General comments

  1. Authors should seek native English speaker to improve the quality or use MDPI language service. I have not attempted to correct English, except where the meaning is unclear.
  2. Referencing has been completely mixed (probably authors have used manual referencing rather than computer-assisted) – References should start from 1 in the text, not 26 (that is most likely incorrect anyway). I did not attempt to correct the quality of referencing, as the current version do not allow for this.  Authors should carefully check all references and correct in the manuscript before re-submission.
  3. The title is not informative enough. Please try something like ‘Diagnostic classification of cases of canine leishmaniasis using machine learning’
  4. In your manuscript you talk about ‘dogs’. Do not generalise your writing to ‘animals’.  Throughout the manuscript, please

Specific comments

L21 – on 340 animals (pleas add n of positive and negative cases)

L38 – ‘affirm’ – word choice

L70 – FTIR – abbreviation not explained

L92 – 94 ‘… on physical examination.  For this purpose, we take some data from canine exams carried out in certain regions of the state of Maranhao, being with low Human Development Index (HDI) [18]. (Figure 1) …’ Note reference may need correction.

L96 – 98 Please delete.

L107- 108 – ‘… comprises professional assessment of the patient’s health and wellbeing.’

L109 – 111 – ‘… database originated from existing clinical examination records on 340 dogs.’

L112 ‘the specialists’ – word choice

L120 – 122 Please delete (addressed above – recommendation related to L 92 – 94)

Table 1 – Augmented – word choice; e.g., for lymph nodes is probably ‘enlarged’

Tale 1 – Presence of skin lesions – the type of skin lesion seen should be described in footnote of the table.  Note that authors already present some secondary skin lesions below this (e.g., alopecia, depigmentation)

L208 – ‘condition’ – word choice, try ‘list’

L210 - ‘and it is common’ should become ‘and are common’

L223 and L226 - Sensitivity and Specificity are mixed up.  Please correct.

L226 – ‘realize’ – word choice – try ‘carry out’

L270 – ‘based’ not ‘base’

L271 – 272 ‘… enables cost reduction in this type of care and can become a useful tool to screen for this disease, ultimately contributing …’

Author Response

(The authors gave the same response as above.)

Round 2

Reviewer 1 Report

The authors didn't address important points from the previous review. Model comparison is an important part of any model-building process. The authors have a single model with 17 variables, and they do not describe any variable selection procedure nor a comparison with different combination of variables, they only include a list of p-values, where it is clear that some coefficients are not statistically significant, but it's hard to infer more from it since they do not include the values or any descriptive information. Answers to review points 3) and 4) are not satisfactory either, and the use of the English language can still be improved.

Author Response

Dear Reviewer,

Thank you so much for your review.

The authors didn't address important points from the previous review. Model comparison is an important part of any model-building process.

Reply: we tested more three models: Naïve Bayes, K-nearest neighbor, and Support Vector machines classifiers.

 The authors have a single model with 17 variables, and they do not describe any variable selection procedure nor a comparison with different combination of variables, they only include a list of p-values, where it is clear that some coefficients are not statistically significant, but it's hard to infer more from it since they do not include the values or any descriptive information.

Reply: we included a selection variable step for each model tested, as described in the manuscript.

Answers to review points 3) and 4) are not satisfactory either.

Reply: We included tables with the descriptive analysis of each variable for each classification class. We included the parameter vector a for the logistic regression model and an explanation of how to classify an unknown sample.

Reviewer 2 Report

The authors took into account my comments, but my previous opinion is still critical. The proposed method can be applied to any data and we will
always achieve a better or worse classification.
The submitted work is of an application nature without any news.
We cannot be sure that for other classifiers (other than logistic regression)
we will not obtain better performance (diagnostics). I am asking for a more in-depth justification of
the chosen method and justification for excluding other,
numerous classification methods.

Author Response

Dear Reviewer,

Thank you so much for your review.

Comments and Suggestions for Authors

The authors took into account my comments, but my previous opinion is still critical. The proposed method can be applied to any data and we will always achieve a better or worse classification. The submitted work is of an applied nature without any news. We cannot be sure that for other classifiers (other than logistic regression) we will not obtain better performance (diagnostics). I am asking for a more in-depth justification of the chosen method and justification for excluding other, numerous classification methods.

Reply: we tested more three models: Naïve Bayes, K-nearest neighbor, and Support Vector machines classifiers and selection variables pre-processing procedures. We also included tables with the descriptive analysis of each variable for each classification class. We included the parameter vector a for the logistic regression model and an explanation of how to classify an unknown sample. We also included tables with descriptive analysis of the results for each classification (TN, FP, FN, TP) for the best model.
